# Radioprotective effect of the anti-diabetic drug metformin

**Silvia Siteni**[ID]°, **Summer Barron**°, **Krishna Luitel**°, **Jerry W. Shay**[ID]*

Department of Cell Biology, University of Texas Southwestern Medical Center, Dallas, Texas, United States of America

☯ These authors contributed equally to this work.
* jerry.shay@utsouthwestern.edu

**Data Availability Statement:** https://doi.org/10.18738/T8/JPAKKV.

**Funding:** This work was supported by NASA Grants # NNX16AE08G to JWS and NNX15AI21G to J.W.S and Albert Fornace. K.L. was supported by Cancer Training Grant T32CA124334. This work

## Abstract

Metformin is a biguanide currently used in the treatment of diabetes mellitus type 2. Besides its anti-glycemic effects, metformin has been reported to induce different cellular pleiotropic effects, depending on concentration and time of treatment. Here we report one administration of metformin (0.5 mM) has radioprotective effects *in vitro* on BJ human fibroblasts, increasing DNA damage repair and increasing SOD1 expression in the nucleus. Importantly, metformin (200 mg/kg) pre-administration for only 3 days in wild type 129/sv mice, decreases the formation of micronuclei in bone marrow cells and DNA damage in colon and lung tissues compared to control irradiated mice at sub-lethal and lethal doses, increasing the overall survival fraction by 37% after 10Gy total body irradiation. We next pre-treated with metformin and then exposed 129/sv mice, to a galactic cosmic rays simulation (GCRsim), at the NASA Space Radiation Laboratory (NSRL). We found metformin pre-treatment decreases the presence of bone marrow micronuclei and DNA damage in colon and lung tissues and an increase of 8-oxoguanine DNA glycosylase-1 (OGG1) expression. Our data highlight a radioprotective effect of metformin through an indirect modulation of the gene expression involved in the cellular detoxification rather than its effects on mitochondria.

## Introduction

Ionizing radiation (IR) induces direct damage on cellular DNA, lipids and proteins as well as indirectly through the radiolysis processes [1]. The radiolysis of water molecules generates reactive oxygen species (ROS), hydroxyl radicals ($^{\bullet}OH$), ionized water ($H_2O^+$), and hydrogen radicals ($H^{\bullet}$), whereas superoxide ($O_2^{\bullet-}$) and hydrogen peroxide ($H_2O_2$) are formed as secondary ROS products of IR [2]. Low-dose radiation is an important diagnostic tool (e.g., computed tomography scanning) while high-dose radiation is a well-known approach for the treatment of many tumor types. Although most of the IR sources are human created, the Sun is the biggest source of radiation on Earth. Giant explosions on the solar surface are responsible for large amounts of x-rays, gamma rays, protons and electrons. Another source of radiation in space is due to galactic cosmic radiation (GCR). GCR consists of heavy, high-energy ions of

was performed in laboratories constructed with support from NIH grant C06 RR30414. We also acknowledge the Harold Simmons NCI Designated Comprehensive Cancer Center Support Grant (CA142543).

**Competing interests:** J.W.S. holds the distinguished Southland Financial Corporation chair in Geriatrics Research. J.W.S is a SAB member of Reata Pharmaceutical, Inc. (Irving, TX). S.S, S.B and K.L. declare no competing interests.

elements that have had all their electrons stripped away as they journey through the galaxy at nearly the speed of light. Life on Earth is partially protected from solar flares and GCR, but GCR ions pass through spacecraft and the skin of astronauts [3]. Dosimetry on unmanned missions in deep space demonstrates that the amount of radiation exposures to an astronaut on a round trip mission to Mars will exceed the safe limits established by NASA that is set to the value of 600mSv. Thus, investigations into radiation countermeasures are a major ongoing area of investigation.

Metformin is the first line therapy for treatment of diabetes mellitus type 2 (T2D) and it is currently prescribed to over 150 million patients worldwide [4]. Metformin is a biguanide that has been used in Europe for treatment of hypoglycemia since 1957 and was FDA approved in 1994. The drug is well tolerated; it does not induce hypoglycemia or weight gain and ameliorates hyperglycemia with remarkable cardiovascular safety. Although its mechanism of action is still elusive, metformin has antioxidant effects, mainly targeting mitochondria, inhibiting the NADH: ubiquinone oxidoreductase (complex I) [5]. The consequent increase in both ADP/ATP and AMP/ATP ratios, determines the activation of the AMP-activated protein kinase (AMPK), that has pleiotropic effects on cell metabolism [6]. Another antioxidant mechanism of action of metformin is through the upregulation of superoxide dismutase enzymes SOD, involved in the protection from superoxide radicals [7]. Interestingly, metformin has been reported to attenuate DNA damage induced by endogenous levels of reactive oxygen species (ROS) *in vitro* [8] and *in vivo*, and also decreases chronic inflammation [9].

Because of its antioxidant effects, we reasoned that metformin could be a medical countermeasure for protecting first responders to nuclear accidents and well as to protect astronauts space associated radiation exposure. Thus, we investigated metformin as a radioprotector, administered before γ-ray exposure *in vitro* by evaluating antioxidant and DNA damage protection effects on human fibroblasts. We also administrated metformin before total body irradiation (TBI) *in vivo* in wild type 129/sv mice, to expand our knowledge on the molecular mechanisms behind the radioprotective effects. Finally, we evaluated metformin as a radioprotector in mice exposed to GCR, using the GCR simulator at the NASA Space Radiation Laboratory (NSRL) at DOE's Brookhaven National Laboratory.

## Results

### Metformin 0.5mM does not affect long-term growth in normal human BJ fibroblasts, induces AMPK phosphorylation and activates a detoxification mechanism through expression of SOD1

We first performed a Cell Titer Glo (CTG) analysis to study the short-term effects of metformin on human BJs fibroblasts and observed an $IC_{50}$ value of 25 mM (S1A Fig). We then evaluated the expression of the superoxide dismutase SOD1 at the concentration of 0.05, 0.1, 0.5, 1mM after 72 hours. We found 0.5 mM and 1 mM metformin showing the highest expression of SOD1 (S1B Fig). To determine the impact of metformin on long-term survival, we performed a clonogenic assay with normal human BJ fibroblasts. Cells were pre-treated with 0.5, 1, 2.5 and 5 mM metformin one time and seeded at low density. After 30 days, colonies were counted, and plating efficiency evaluated (Fig 1B). Results show that 0.5 mM metformin does not affect the long-term survival and colony formation in normal BJs cells, whereas increasing doses affects cell viability (Fig 1A). Thus, subsequent studies used 0.5 mM metformin.

The positive effects of metformin on hepatic glucose decrease in type 2 diabetes (T2D) are linked to AMPK activation [6]. AMPK is a serine/threonine-protein kinase and consists of a heterotrimeric complex containing a catalytic subunit α and regulatory β and γ

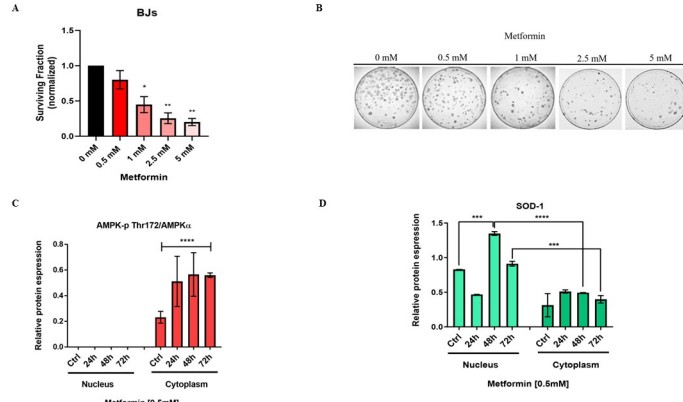

**Fig 1. Metformin 0.5 mM does not affect long-term growth in normal human BJ fibroblasts, induces AMPK phosphorylation and activates a detoxification mechanism through expression of SOD1.** (A) Histogram showing the surviving fraction of BJs treated with 0.5, 1, 2.5 and 5 mM metformin. (B) Representative colony forming unit assay of BJ cells, at different metformin concentration (unpaired t-test, *p<0.05, **p<0.01). (C) AMPK-p Thr 172 and (D) SOD1 expression in nucleus and cytoplasm of BJ fibroblasts control and treated with metformin 0.5 mM at 24,48 and 72 hours, with SOD1 increasing the expression in the nucleus after 48 hours treatment (two-way ANOVA, ***p<0.001, ****p<0.0001).

subunits. The phosphorylation of the α subunit at the threonine residue (Thr 172) by upstream kinases is necessary to activate AMPK and decrease energy metabolism [10–12]. Superoxide dismutases (SODs) are a group of metalloenzymes participating in ROS removal, catalyzing the dismutation of superoxide into oxygen and $H_2O_2$ [13]. Although there are three different superoxide dismutases, SOD1, SOD2 and SOD3 in eukaryotic cells, only SOD1 has been reported to localize in the nucleus in response to elevated $H_2O_2$ by binding to gene promoters involved in oxidative stress, replication stress, and DNA damage responses [14]. We first assessed if 0.5 mM metformin results in the phosphorylation of sub-unit α of AMPK Thr 172 in BJs fibroblasts. Metformin 0.5 mM resulted in a significant increase in AMPK-p Thr 172 expression in the cytoplasm in the first 72 hours (Fig 1C, S1C Fig). We found that metformin also significantly increased the expression of SOD1 in the nucleus compared to the control of the nuclear fraction as well as in the cytoplasm, after 48 hours (Fig 1D, S1C Fig). This indicated that one administration of 0.5 mM metformin induces AMPK-p Thr 172 expression and activates a mechanism of detoxification via SOD1 in human normal BJ fibroblasts.

## SOD1 expression is dependent on AMPK phosphorylation at residue Thr172

Next, we assessed if SOD1 increased expression in the nucleus at 48 hours is related to AMPK activation. Compound C (6-[4-(2-Piperidin-1-ylethoxy) phenyl]-3-pyridin-4-ylpyrazolo [1,5-a]pyrimidine) is a cell-permeable inhibitor of AMPK [15, 16]. BJs cells were treated with 0.5 mM metformin for 48 hours with or without Compound C (CC) 35 µM for 24 hours (Fig 2A, S2A Fig). As expected, CC inhibited the expression of AMPK (Fig 2B and 2C) and we did not observe any increase in SOD1 expression treated with both metformin and CC, whereas treatment with metformin only, significantly increased the expression of SOD1 in the nucleus (Fig 2B and 2D). We conclude that the detoxification in BJs cells via SOD1 expression, is dependent on the activation of AMPK-p Thr 172, in the presence of metformin.

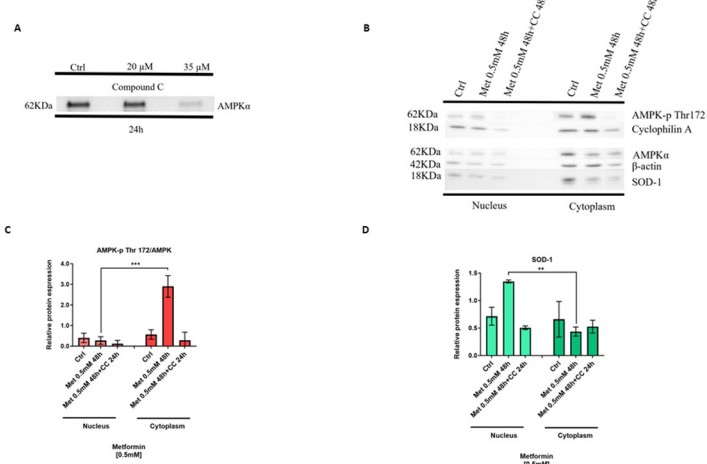

**Fig 2. SOD1 expression is dependent on AMPK phosphorylation at residue Thr172.** (A Western blot showing compound C (CC) 35μM effectively inhibits AMPK expression in BJ fibroblasts. (B) Compound C administrated 24 hours after metformin 0.5 mM 48 hours treatment, inhibits AMPK-p Thr 172 expression and affect the expression of SOD1 in the nucleus. (C) AMPK-p Thr 172 and (D) SOD1 expression significantly decrease in presence of CC treatment, suggesting metformin increases the expression of SOD1 in the nucleus, dependent on the phosphorylation of AMPK (two-way ANOVA, ***p<0.001, ****p<0.0001). Original blots.

## Metformin determines mitochondrial membrane hyperpolarization during the first 30 minutes of treatment and shows radioprotective effects *in vitro*

Metformin inhibits the mitochondrial respiratory-chain complex I [17], determining membrane depolarization [18] with consequent release of ROS. To investigate the effect of metformin on mitochondria membrane changes, we treated BJs cells with metformin and performed a tetramethylrhodamine ethyl ester (TMRE) assay. We found that metformin induced a hyperpolarization of the mitochondria membrane in the first 30 minutes treatment, compared to the control (Fig 3A). Interestingly, we did not observe any change in ROS endogenous level during the first 60 minutes of treatment (Fig 3B).

To evaluate possible radioprotective activity of metformin, we irradiated BJs cells with 2 or 4 Gy of γ-rays, control or pre-treated with 0.5 mM metformin for 48 hours. Then, we evaluated the number of DNA double strand breaks via γH2AX foci quantitation (S3a Fig). There was a significant decrease in the number of foci when BJs cells were pre-treated with metformin. This suggests metformin repairs faster or mitigates DNA damage induced by ionizing radiation (IR) (Fig 3C). To supplement these results, we performed a Comet assay (Fig 3D), and observed a significant decrease in the tail DNA percentage, in BJs pre-treated with metformin, compared to the irradiated controls after 2 and 4 Gy irradiation (S3B Fig).

## Pre-administration of metformin 3 days before irradiation significantly decreases the number of micronuclei in murine bone marrow cells and the number of DNA damage foci in colon and lung tissues, increasing mice survival after exposure to ionizing radiations

To investigate the possible radioprotective effect of metformin *in vivo*, wild type 129/sv mice were pre-treated with metformin 200 mg/kg via i.p. for three consecutive days, irradiated at the sub-lethal, whole-body dose of 7.5 Gy of X-rays, and then the number of micronuclei (MN) in bone marrow cells was evaluated. MN can be derived from acentric fragments not

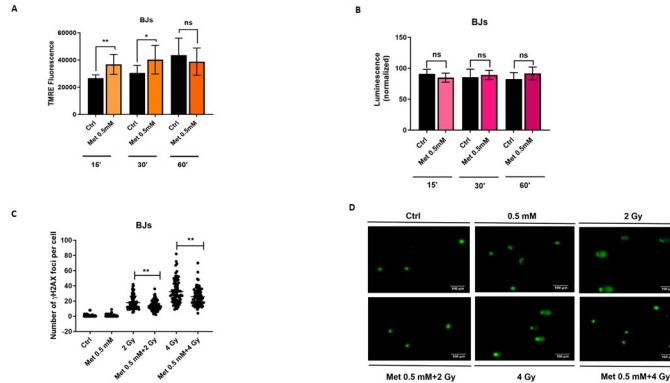

**Fig 3. Metformin determines mitochondrial membrane hyperpolarization during the first 30 Minutes of treatment and shows radioprotective effects *in vitro*.** (A) Tetramethylrhodamine ethyl ester (TMRE) assay performed on BJ fibroblasts in presence or absence of metformin 0.5 mM at 15, 30 and 60 minutes. Metformin 0.5 mM induces a transient hyperpolarization during the first 30 minutes of treatment (student's t-test, Welch correction, *p<0.05, **p<0.01). (B) Reactive oxygen species (ROS) detection assay shows no difference in ROS endogenous production in presence of metformin 0.5 mM, during the first 60 minutes of treatment. (C) Pre-treatment of BJ fibroblasts with metformin 0.5 mM and irradiated with 2 or 4 Gy γ-rays, after 48 hours, show a lower number of γH2AX DNA damage foci and of the (D) tail DNA percentage, compared to the irradiated controls (student's t-test, Welch correction, *p<0.05, **p<0.01).

repaired during a DNA insult, or from lagging chromosomes during mitosis [19]. IR can induce MN formation (among other cellular damages), and thus MN are consider a marker of genotoxicity [20, 21]. We found that pre-administration of metformin significantly decreases the number of micronuclei in bone marrow cells (BMCs), compared to the irradiated controls, decreasing the number of MN by 38% and 57% at 48 and 72 hours, respectively (Fig 4A).

To assess if metformin has radioprotective effects on tissues, wild type 129/sv mice pre-treated with metformin 200 mg/kg via i.p. for three consecutive days, then irradiated at the

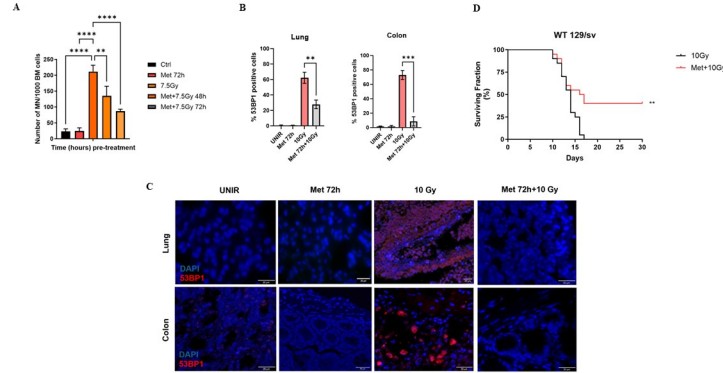

**Fig 4. Pre-administration of metformin 3 days before irradiation significantly decreases the number of micronuclei in murine bone marrow cells and the number of DNA damage foci in colon and lung tissues, increasing mice survival after exposure to ionizing radiations.** (A) Metformin 200 mg/kg pre-administration significantly decreases the number of micronuclei in mice bone marrow cells, compared to the 7.5 cGy control, in 129/sv mice one-way ANOVA, **p<0.01, ****p<0.0001). (B, C) Pre-administration of metformin 200 mg/kg in mice for three consecutive days, decreases the number of 53BP1 foci of mice irradiated at the lethal dose of 10 Gy (one-way ANOVA, **p<0.01, ***p<0.001). (D) Long-term survival of wild type 129/sv mice feed with metformin chow 200 mg/kg for three consecutive days and then irradiated at the lethal dose of 10 Gy of X-rays (log-rank test, Chi square 7.1, **p<0.01).

lethal dose of 10 Gy of X-rays. After 24 hours, we analyzed the number of 53BP1 foci in colon and lung tissues, as markers of double stranded DNA damage [22, 23]. The number of 53BP1 positive cells was significantly less in the colon and lung tissues of mice pre-treated with metformin, compared to the irradiated control. The number of 53BP1 positive cells was ~60% and 30% less, in colon and lung tissue respectively, compared to the irradiated control (Fig 4B and 4C), showing metformin mitigates DNA damage induced by X-rays *in vivo*.

To determine the effect of metformin on long-term survival, wild type 129/sv mice were fed metformin chow 200 mg/kg for three consecutive days and then irradiated at the lethal dose of 10 Gy of X-rays. Overall survival was monitored for 30 days post irradiation. Surprisingly, 37% of mice pre-treated with metformin survived, whereas all irradiated control mice died during the first 16 days (Fig 4D).

## Pre-administration of metformin significantly reduces the number of micronuclei in bone marrow cells and mitigates DNA damage in colon and lung tissues, after exposure to the 75 cGy 33-beam GCRsim irradiated mice

Thanks to the NSRL at Brookhaven National Laboratory, it is now possible to simulate galactic cosmic radiation (GCR) that occurs in space, allowing investigators to more closely determine the effects of deep space irradiation on biological models. NASA has developed the capability to simulate primary and secondary galactic cosmic radiation (GCRsim) using 33 ion-energy fast switching combinations to more closely simulate human space exploration [24, 25]. The GCR spectrum is composed of 33 beams, uses mixed ion species, protons and sporadic heavy ions to better simulate the space environment [26]. Furthermore, it is possible to investigate different dose-rate effects. Protons account for nearly 87% of the total flux, helium ions account for approximately 12%, and the remaining heavy ions account for less than 1% of the total flux [24]. A Mars mission of 650–920 days is about 300–450 mGy, thus a Gy equivalent of 550–800. Relevant GCR doses are 125, 250, 500, and 750 mGy for radiobiology experiments at the NSRL [27].

To evaluate the genotoxic protection of metformin from GCRsim, wild type mice were fed with 200 mg/kg metformin chow for 72 hours and then irradiated with the Mars mission dose of 75 cGy GCRsim. After 4 days, a subset of mice was sacrificed to perform micronuclei assays of bone marrow cells. As a pilot study, we found that 75 cGy acute dose significantly increases the number of micronuclei compared to the unirradiated control and pre-administration of metformin decreases the number of micronuclei induced by GCRsim (Fig 5A).

To investigate DNA damage responses in colon and lung tissues, wild type mice were pre-fed with metformin chow 200 mg/kg for 72 hours and then irradiated with 75 cGy GCRsim. Mice were sacrificed 6 hours post-irradiation, and colon and lung tissues stained with 53BP1 to analyze the number of foci (Fig 5B). Both tissues showed a 2-fold decrease in the number of 53BP1 positive cells, compared to the irradiated control, suggesting a radioprotective effect of metformin from GCRsim (Fig 5C).

## Metformin retains OGG1 glycosylase enzyme expression and protects from GCRsim-induced apoptosis in colon and lung tissue

To investigate the effects of GCRsim on oxidative stress, we evaluated 8-oxoguanine (8-oxoG) as a biomarker of oxidative stress [28–30]. 8-oxoG is formed by the oxidation of a guanine base in the DNA and the OGG1 is able to specially remove 8-oxoG [31]. Wild type mice were fed with metformin for 72 hours and then irradiated with 75 cGy GCRsim. After 4 days, we analyzed the expression of OGG1 in colon and lung tissues. Pre-treatment with metformin significantly retains the expression of OGG1 in colon and in lung tissue (Fig 6A and 6B, S4A and

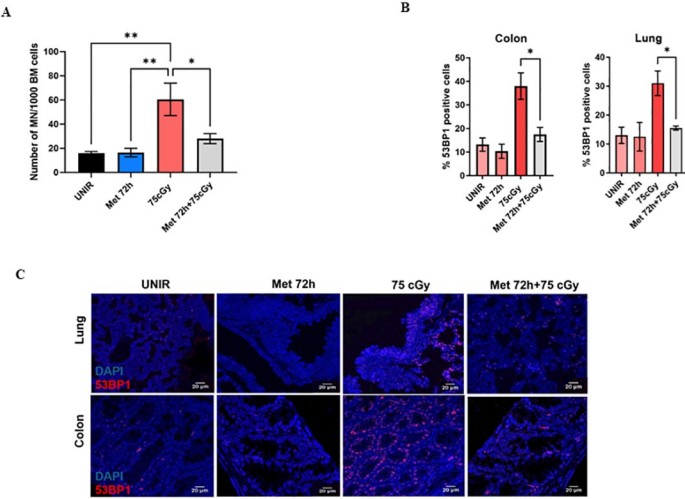

**Fig 5. Pre-administration of metformin significantly reduces the number of micronuclei in bone marrow cells and mitigates DNA damage in colon and lung tissues, after exposure to the 75 cGy 33-beam GCRsim irradiated mice.** (A) Pre-administration of metformin chow 200 mg/kg decreases the number of MN in murine bone marrow cells, GCRsim irradiated. (B, C) Metformin pre-treatment decreases the number of 53BP1 positive cells significantly in colon and lung tissues of irradiated wild type mice (one-way ANOVA, *p<0.5, **p<0.01).

S4B Fig). We also evaluated the expression levels of cleaved PARP, as a marker of apoptosis [32]. Irradiation with 75 cGy GCRsim significantly increases the cleaved-PARP expression compared to the unirradiated controls in colon but not in lung tissue. In addition, metformin pre-treatment retains the expression levels of cleaved-PARP similarly to the control (Fig 6C and 6D, S4a and S4b Fig). Finally, we found that 72 hours metformin pre-treatment activates the phosphorylation of AMPK, suggesting a main role in the radioprotective pathway (Fig 6e and 6f, S4a and S4b Fig).

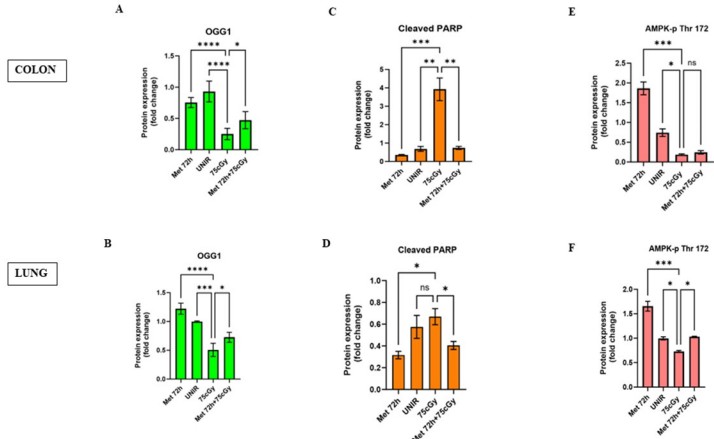

**Fig 6. Metformin pre-treatment retains an antioxidant effect and decreases apoptosis, in an AMPK activation manner.** (A, B) Pre-administration of metformin chow 200 mg/kg in 129/sv mice, retains the expression of OGG1. (C, D) Furthermore, metformin significantly decreases the expression of cleaved PARP, compared to the irradiated control. (E, F) Metformin activates AMPK through the phosphorylation at Thr 172 (One-way ANOVA, *p<0.05, **p<0.01, ***p<0.001, **** p<0.0001).

## Discussion

Metformin remains the preferred first-line pharmacologic treatment for type 2 diabetes. However, metformin has been reported to have additional applications, including as a radioprotective compound [33]. One of the main issues about metformin is the usage of the correct concentration that is safe. Supra-pharmacological concentrations of metformin are not translatable in human beings, and, importantly pharmacological or supra-pharmacological concentrations lead to different cellular effects [34]. We demonstrated 0.5 mM metformin does not affect the long-term clonogenic or survival of normal diploid BJ fibroblasts. Metformin activates AMPK through the phosphorylation of the AMPK catalytic α subunit at Thr172 in primary hepatocytes [6], with consequent pleiotropic effects on cell metabolism [10–12]. We observed a significant increase in AMPK-p Thr172 expression in the first 72 hours treatment with 0.5 mM metformin in BJ human fibroblasts. We also confirmed the presence of AMPK-p Thr172 in the cytoplasm fraction only, excluding AMPK translocation in the nucleus, as reported elsewhere [35]. SOD1 is an anti-oxidant enzyme responsible for the dismutation of superoxide anions and it is associated with the expression of antioxidant and repair genes, acting as a nuclear transcription factor to regulate oxidative stress resistance [14]. Furthermore, overexpression of SOD1 correlates with radioresistance in human glioma cells [36]. We observe SOD1 expression increases in the nucleus 48 hours after treatment with 0.5 mM metformin, suggesting 0.5 mM metformin activates a detoxification mechanism in human BJ fibroblasts. Interestingly, Kaneb et al. [37] reported metformin treatment did not show any beneficial effect in the SOD1G93A mouse model of Amyotrophic Lateral Sclerosis (ALS). However, metformin has been shown to attenuate the pathology in mouse models of Huntington's disease and multiple sclerosis [38, 39], suggesting the need of a functional SOD1 for metformin to exert an anti-oxidant effect. We also inhibited AMPK activation, treating BJ cells with the AMPK inhibitor compound C. The inhibition of AMPK activation with compound C, shows a concomitant inhibition of SOD1 expression in the nucleus, suggesting an anti-oxidant activity of metformin is dependent on the activation of AMPK.

Metformin has been reported to act directly on mitochondria to inhibit complex I-mediated mitochondrial respiration, resulting in mitochondria membrane depolarization and an energetically inefficient mitochondria metabolism [18]. We found that in BJ fibroblasts, 0.5 mM metformin does not change mitochondrial membrane polarization (MMP), compared to the control at 24, 48 and 72 hours, but we observed an increase in the mitochondrial membrane potential in the first 30 minutes with 0.5 mM metformin treatment in BJ fibroblasts. This is in accordance with what was reported by Wang et al.[40] who showed that treatment with supra-pharmacological metformin concentrations (500 and 1,000 μM) do not reduce mitochondrial membrane potential in primary hepatocytes. However, the treatment with 1,000 μM metformin significantly increased mitochondrial membrane potential in other cell lines. We also did not find a decrease of ROS during the first 60 minutes of treatment with 0.5 mM metformin, suggesting that metformin does not act on mitochondria to decrease the endogenous level of ROS.

At the present time ionizing radiations are part of our lives for many useful applications (e.g. nuclear power generation and diagnostic medical procedures), but they might be also involved in more dangerous situations such as accidental exposures or terroristic attacks. Furthermore, NASA is concerned about astronaut's radiations exposure for future long-term missions to the Moon and Mars. BJ fibroblasts exposed to 2 and 4 Gy ionizing radiation show a decrease in γH2AX foci when treated with 0.5 mM metformin 48 hours prior to γ-rays exposure. We also observed similar results during the alkaline comet assay, with a significant decrease of DNA tail percentage in cells pre-treated with metformin. Our data can be

interpreted to suggest that pre-treatment with metformin may improve DNA repair kinetics in irradiated BJ cells. Metformin 0.5 mM does not affect the long-term viability in BJ fibroblasts and 200 mg/kg metformin does not increase the expression of cleaved-PARP in mice, suggesting that metformin does not increase the fraction of apoptotic cells. This is consistent with other reports showing metformin has anti-apoptotic effects in rat hepatocytes [41, 42], tubular cells [43] and myocardiocytes [44]. A possible role of the radioprotective effect of metformin might involve the ATM kinase that acts upstream of p53 and controls a DDR pathway critical to resolving DSBs [45]. Vazquez-Martin et al. [46] reported ATM phosphorylation at serine 1981 and formation of γH2AX foci after metformin exposure in epidermoid carcinoma cells, but not of 53BP1. This is in line with what reported by Bakkenist & Kastan [47], showing ATM activation is not dependent on direct binding to DNA strand breaks, but may result from changes in the structure of chromatin. Interestingly, the increased expression of p53 might have effects on the cell cycle, inducing a transient arrest in G1 phase, and thus protecting the DNA from the effects of IRs.

*In vivo*, pre-administration of 200 mg/kg metformin for three days showed a decrease in the number of micronuclei (MN) in bone marrow cells after 7.5 Gy total body irradiation, compared to the 7.5 Gy irradiated control mice. These results show the ability of metformin to decrease the genotoxic effects induced by IRs, is consistent with what reported by Dan et al. [48] and Cheki et al. [33]. Moreover, we found a decrease of 53BP1 positive cells, in lung and colon tissues of mice treated with metformin before providing the total body lethal dose of 10 Gy X-rays. Interestingly, we found the most effective results on colon tissue (where metformin is absorbed) was likely linked to a decrease level of pro-inflammatory cytokines [48].

We then showed the effects of metformin pre-treatment on mice survival. We pre-administered 200 mg/kg metformin chow for 3 days and then mice were irradiated at the lethal dose of 10 Gy X-rays. Surprisingly, metformin increased by 37% the survival fraction compared to the irradiated control mice. We also administrated 200 mg/kg metformin chow after 10 Gy exposure, to investigate metformin mitigating effects, but we did not observe any difference in the mice survival compared to the irradiated control (data not shown). We interpret these results to suggest metformin acts on cellular metabolism to protect mice from external injuries, such as high dose of IRs. This is consistent with our *in vitro* data, showing a change in cell metabolism, by increasing SOD1 nuclear expression, rather than endogenous ROS decreases induced by metformin treatment. Even though we did not observe metformin (200 mg/kg) being effective as a radiomitigator, when provided at the time or following lethal doses of IRs, others reported metformin can activate different pathways at lower concentrations [40]. When 200 mg/kg metformin was administrated immediately after a sublethal dose of radiation, radiomitigating effects on BALB/c mice [48] was observed.

For long-term astronaut space missions to the Moon or Mars, it might be necessary to adopt safe and oral available radiological countermeasures since the amount of radiation exposure predicted for a round trip to Mars currently exceed the limits for safe flying days. Radiation shielding is an effective countermeasure for solar particle events (SPEs), but the energy spectrum of the galactic cosmic rays (GCRs) peaks near 1 GeV/n, such that particles are so penetrating that shielding can only partially reduce the doses absorbed by the crew [49].

Thanks to the NSRL at DOE's Brookhaven National Laboratory, we were able to study the effect of galactic cosmic radiation simulations (GCRsim) on mice and the possible effect of metformin pre-administration. Metformin 200 mg/kg chow provided three days prior to 75 cGy GCRsim acute dose showed a significant decrease of MN in bone marrow cells as well as a decrease in the number of 53BP1 positive cells, compared to the irradiated control 4 days post-irradiation.

To evaluate the antioxidant activity of metformin, we analyzed the expression of 8-oxoguanine DNA glycosylase 1 (OGG1), the most important enzyme involved in the repair of 8-oxoguanine (8-oxoG), a base modification induced by ROS. Pre-administration of metformin increased the expression of OGG1 in lung but not in colon tissue. This can be explained with the absorption and pharmacokinetics of metformin in different tissues. Metformin is absorbed in the intestine [50] where it accumulates at much higher concentrations than in plasma [51], with a concentration peak in lung below the 2% [52]. Metformin pre-administration does not increase the expression of cleaved-PARP, suggesting that radioprotective activity of metformin is not related to an increase in the apoptotic cells fraction. Finally, we found the activation of AMPK to be related to the OGG1 activity and to the decrease of cleaved-PARP in pre-treated mice, confirming the AMPK activation role in the antioxidant and radioprotective activity of metformin. The present results, along with previous studies by others, suggest that metformin is an excellent radioprotector and may have important medical applications for first responders to nuclear accidents, astronauts on long-term missions in space, and potentially to reduce side effect for patients receiving radiation therapy.

## Materials and methods

### Cell lines

BJ human foreskin fibroblasts were obtained from the ATCC (Manassas, VA) and grown in Medium X (DMEM:199, 4:1, Hyclone, Logan, UT) supplemented with 10% cosmic calf serum (Hyclone, Logan, UT) without antibiotics and incubated in a humidified atmosphere with 5% $CO_2$ at 37°C. Only BJs with a population doubling between 5 and 15 were used. BJ cells are very stable in vitro maintaining a normal karyotype for over 50 population doubles. Thus, the BJ cells used in the present studies would still be considered young or middle aged and not old.

For irradiation experiments, BJ fibroblasts were grown in a 10 mm dishes and treated with different concentration of metformin only one time. After three days, plates were irradiated with 2 or 4 Gy of γ-rays.

BJ fibroblasts were irradiated with γ-radiation using a 137Cs source at a dose rate of 243.08 cGy/min at UTSW. Dosimetry of the irradiator was carefully monitored for accuracy by radiation physicists in the Department of Radiation Oncology at UT Southwestern Medical Center.

### Drugs preparation

For *in vitro* studies, metformin hydrochloride (Sigma Aldrich) was dissolved in nuclease free water (Ambion) to prepare 1M stock solutions, which were kept frozen at −20°C and diluted at different concentrations according to the experiment.

For mouse *in vivo* studies, metformin hydrochloride was prepared in nuclease free water (Ambion) to prepare 1M stock solutions for intraperitoneal (i.p.) injection. Metformin chow (ENVIGO) was made at 1000 ppm, equal to 200 mg/kg concentration. Metformin administration in human is between 1000 and 2000 mg per day, and the human equivalent dose (HED) is 20–30 mg/kg. For a more appropriate conversion of drug doses from animal studies to human studies, we used the body surface area (BSA) normalization method [53]:

$$\text{Animal dose} = \text{Human } K_m / \text{Animal } K_m \text{ x HED}$$

Where human $K_m$ and animal $K_m$ are respectively 37 and 3, according to FDA guidelines [54] and thus for a minimal dosage of 250 mg/kg. However, since our aim was to study the radioprotective effects of metformin, we decreased the concentration to 200 mg/kg.

Compound C (Sigma Aldrich) was reconstituted in 2 mg/ml DMSO (Sigma Aldrich) and then diluted in nuclease free water (Ambion) at the concentration of 10 mM. Aliquots were stored at -20˚C.

## Cell viability assay

For $IC_{50}$ determination, BJ cells were screened with metformin hydrochloride or compound C with a four-fold dilution series in 8 different points in 96-well plates. Cells were plated 24 hours prior to addition of drug, incubated for 3 days, and assayed for CellTiter-Glo luminescent cell viability assay according to the manufacturer's instructions (Promega). Cell number was 10000 cells per well. Dose-response curves were generated and $IC_{50}$ calculated using Graphpad Prism. All samples were analyzed in triplicate, and standard deviations are from two independent experiments.

## Colony formation assay

BJ fibroblasts were seeded at 100, 200, 400, 600, 800 and 1000 cell density, in 10 mm Petri dishes, in triplicate. Cells were left to adhere overnight (ON) and then treated one time with 0.5, 1, 2.5, and 5 mM metformin respectively. After 40 days, dishes were washed with PBS 1X and stained with a mixture of 6.0% glutaraldehyde and 0.5% crystal violet, ON. Plates were then carefully rinsed with tap water and air dried at RT. Colonies were counted with a colony counter pen (VWR, Radnor PA). Experiments were repeated three times. Surviving fraction was calculated according to the equation SF = colonies counted/ colonies seeded X PE, where PE is the plating efficiency.

## Immunoblot

Human BJ fibroblasts or mice colon and lung tissues were collected, and protein extracted with a Nuclear Extraction Kit (ab113474, Abcam) according to the manufacture. Protein concentrations were quantified with the Pierce BCA Protein Assay Kit (Thermo Fisher Scientific), using a PHERAStar FS plates reader (BMG Labtech) in the absorbance range of 540 nm and 590 nm. Each sample was analyzed in triplicate. For western blot assay, 40μg of protein extracts were loaded onto a Mini-PROTEAN TGX PRECAST Gel (Cat. 4561086, Bio-Rad,) and ran at 100V for 30 minutes, in 1X Tris/Glycine/SDS buffer (Cat. 1610731, Bio-Rad). After transferring protein to a PVDF membrane using a Trans-Blot Turbo Transfer Pack (Cat. 1704157, Bio-Rad), blocking buffer (5% dry milk in PBST) was used for 1 h and primary and secondary immunostaining was performed as described previously [55]. The following antibodies were used: AMPK (Cat. 2532, Cell Signaling), phospho-AMPKα (Thr172) (Cat. 2535, Cell Signaling), SOD1 (Cat. sc58421, Santa Cruz), cyclophilin A (Cat. 2175, Cell Signaling), anti β-actin (Cat. ab8227, Abcam), OGG1 (Cat. ab124741, Abcam), cleaved-PARP (Cat. sc56196, Santa Cruz). Pictures were acquired with a G:BOX Chemi Gel Imaging Western Blot Pred Chemi XRQ (Syngene, USA). Bands were quantified with ImageJ.

## TMRE-mitochondrial membrane potential and ROS-Glo $H_2O_2$ assays

TMRE was performed with a TMRE-Mitochondrial Membrane Potential Assay Kit (Cat. ab113852, Abcam) according to the manufactory. 10,000 cells per well were plated in a Costar Assay Plate, 96 well with clear flat bottom (Cat. 3903, Corning) well, treated with metformin 0.5 mM for different time end-points. After drug treatment, TMRE or ROS-Glo detection solution (Promega), were added. Plates were analyzed at Ex/Em 549/575 nm or with

luminescence respectively, with a PHERAStar FS plates reader (BMG Labtech). Each sample was in 8 different time points. Standard deviations were from two independent experiments.

## Cells immunofluorescence

DNA damage foci quantitation was performed as previously described [56]. BJ fibroblasts were seeded onto a slide (Cat. 22035900, Fisherbrand) previously sterilized with ethanol 100% and placed in a 10 cm Petri dish (Falcon). After 24 hours, cells were treated with 0.5 mM metformin, for 48 hours and slides were irradiated with 2 or 4 Gy γ-rays. After 4 hours, slides were rinsed in PBS 1X for 10 minutes on a shaking platform. Slides were then fixed in 4% formaldehyde (Thermo Fisher) for 10 minutes on ice and washed in PBS 1X for 5 minutes, two times. Subsequently, slides were permeabilized with 0.5% Triton X-100 (Sigma Aldrich) for 10 minutes on ice and then blocked with BSA/PBS 1X for 30 minutes at RT. Anti-mouse primary antibody γH2AX (Millipore) was diluted 1:200 in blocking solution and cells incubated in a humid chamber at 4°C ON. Following washes with 1X PBS, cells were incubated with Alexa-Fluor 488 conjugated goat anti-mouse, for 45 minutes at RT. After washes with PBS, slides were counterstained with DAPI/Vectashield (Vector Laboratories). Images were captured at 40X magnification with an Axiovert 200M (Carl Zeiss) equipped with an x-cite 120Q and analyzed with ImageJ software.

## Comet assay

The comet assay for DNA damage evaluation was performed using a comet assay kit (ab238544, Abcam) according to the manufacturer. BJ fibroblasts were seeded and after 24 hours, treated with 0.5 mM metformin for 48 hours and slides were irradiated with 2 or 4 Gy γ-rays. Comet agarose was dropped onto the comet slide to form a base, cells were collected 4 hours post irradiation, combined with comet agarose and dropped on the agarose base. Cells were treated with a lysis buffer (NaCl 14.6 g, EDTA 20 ml, 10X lysis solution 10 ml, DMSO 10 ml, DI water to 90 ml) and an alkaline solution (NaOH 1.2 gr, EDTA 0.2 ml, DI water to 100 ml). Electrophoresis was performed in neutral conditions, in TBE solution (tris base 10.8 g, boric acid 5.5 g, EDTA 0.93 g, DI water to 1 L) at 0.6 V/cm for 20 min. Cells were finally stained and comets detected at 40X magnification with an Axiovert 200M (Carl Zeiss) equipped with an x-cite 120Q and analyzed with ImageJ software. Analysis was performed on 100 cells and data analyzed with the Open Comet plug-in.

## Animal experiments

**Irradiations.** All animal experiments were reviewed and approved by the Institutional Animal Care and Use Committees (IACUC) at the University of Texas Southwestern Medical Center at Dallas (UTSW) and Brookhaven National Laboratory (BNL) (Upton, NY). Wild type 8–12 weeks old 129/sv mice were housed and bred following an approved husbandry protocol in ventilated micro-isolator cages within a pathogen-free facility at UTSW. The irradiation was carried out on a Precision XRAD 320 machine (Precision X-ray Inc, Madison, CT). The x-rays of 250 kVp and 15 mA were applied to the animal cage which was placed on a stainless steel platform at 65 cm from the focal spot. An additional 1.65 mm Al filtration was added to the x-ray beam. The x-ray tube output dose rate was measured using an ionization chamber (PTW 31010) and an electrometer (PTW UnidosE), which was used to obtain the calibration curve of EBT3 gafchromic film (Ashland LLC, Bridgewater, NJ). The film was placed between two pieces of water equivalent materials (each of 1.0 cm thickness) in the animal cage and the dose rate to water was calculated from film measurements with the calibration curve. Small irradiation boxes were used for the housing of mice during radiation exposure, with 2–3 mice

**Table 1.** *In vivo* experimental plan with different radiations sources.

| Metformin pre-treatment | Radiation | Dosage | Time point | Assay |
|---|---|---|---|---|
| 200 mg/kg i.p., 48–72 hours | X-rays | 7.5 Gy | 24 hours after X-rays | Micronuclei |
| 200 mg/kg i.p., 72 hours | X-rays | 10 Gy | 24 hours after X-rays | Tissues IF |
| | | | | Western blot |
| 200 mg/kg chow, 72 hours | X-rays | 10 Gy | 30 days after X-rays | Long term survival |
| 200 mg/kg chow, 72 hours | GCRsim | 75 cGy | 4 days | Micronuclei |
| 200 mg/kg chow, 72 hours | GCRsim | 75 cGy | 6 hours | Tissues IF |
| | | | | Western blot |

fitting comfortably in a single box. These boxes had small holes drilled for the ventilation for the period of radiation exposure. Mice were arranged in the center of the field to assure the best total body irradiation exposure. All GCR mice were placed in small, ventilated holders with cage mates in pairs, and concurrently given an acute 75 cGy whole-body exposure of the NASA Space Radiation Laboratory (NSRL) 33-beam GCR simulation over the duration of 2 hours. Sham-irradiated mice were also placed in small, ventilated holders paired with cage mates, for 2 hours, but did not receive charged-particle radiation. Radiation was delivered in an even 60 x 60 cm beam distribution. Experiments were assessed as reported in Table 1.

**Bone marrow isolation and micronuclei assay.** Wild type 129/sv mice were euthanized with $CO_2$ inhalation and tibias and femurs were flushed into a 50 ml tube through 40 µm cell strainer with Media X to collect bone marrow cells. Cells were centrifuge at 1500 rpm for 8 min and supernatant discarded. Gently, a Carnoy solution fixative (methanol:acetic acid, 3:1) was added drop by drop to the cells until reaching 2 ml. Then, cells were centrifuge at 1500 rpm for 8 min and supernatant discarded. This step was repeated 4 times. After removal of the last supernatant, cells were resuspended in 500 µl-1 ml fixative, depending on the cell density. Cells were then dropped onto a pre-cleaned and wet slide and air dried. Cells were stained with the Differential Quik III Stain Kit (Polysciences) and samples analyzed with an Olympus CX31 upright microscope, at the magnification of 100X. Only white blood cells in interphase were analyzed. A thousand cells were analyzed. Standard deviations were derived from at least 3 different animals per group.

**Tissues immunofluorescence.** Mice were euthanized via $CO_2$ inhalation. Lungs were inflated by intra-tracheal infusion with 4% v/v formaldehyde solution. the trachea clamped. Colons were cut out and cleaned flushing 4% v/v formaldehyde. The whole lung and the colon were immersion-fixed overnight in 4% v/v formaldehyde solution. Tissues were processed, paraffin embedded and cut at 5 microns thick sections. Tissue sections were de-paraffinized in xylene, placed in 100%, then 95% ethanol, and finally rehydrated in deionized water. The tissue sections were then unmasked in a microwave in 10 mM sodium citrate buffer pH 6 at power 5, for 20 minutes then allowed to cool for 30 minutes. Tissue was then blocked with 1X PBS+5% normal serum + 0.3% Triton X-100 at room temperature. Tissues were then incubated with an anti-53BP1 rabbit primary antibody (Novus) overnight. The next day the slides were washed in 1X PBS 3 times and then incubated with AlexaFluor 568 secondary antibody for 1hr in the dark. Finally, the slides were washed with PBS for 5 minutes 3 times then mounting media H-1500 (Vector Laboratories, CA) was added with DAPI and coversliped. Images were captured at 20X magnification with an Axiovert 200M (Carl Zeiss) equipped with an x-cite 120Q and analyzed with ImageJ software.

**Long-term survival.** All procedures and experiments involving mice were approved by The University of Texas Southwestern Institutional Animal Care and Use Committee and conducted as per institutional guidelines. Wild type 129/sv both male and female mice were randomly divided in 3 main group: ctrl (n = 5), 10 Gy X-rays (n = 15), metformin 200 mg/kg plus 10 Gy X-rays (n = 10) and their survival evaluated for 30 days. A Kaplan-Meier curve was established. Standard deviations are relative to 3 different experiments.

**Statistical analysis.** All data analyses and graphs were performed with GraphPad Prism statistical software.

## Supporting information

**S1 Raw images.**
(PDF)

**S1 Fig. Short-term effects of metformin on human BJs fibroblasts.** (A) Cell Titer Glo (CTG) analysis for the short-term effects of metformin on human BJs fibroblasts, IC50 = 25 mM. (B) Western blot showing maximun expression of SOD1 at 0.5 and 1 mM metformin, after one treatment only. (C) Western blot showing the increase expression of AMPK-p Thr 172 after one treatment only with metformin 0.5 mM, during the next 72 hours as well as an increase of SOD1 expression after 48 hours treatment in the nucleus.
(TIF)

**S2 Fig. Compound C inhibits the expression of AMPK.** Cell viability curve in presence of different concentration of Compound C after 24 hours treatment in human BJ fibroblasts, $IC_{50}$ = 35 μM.
(TIF)

**S3 Fig. Metformin shows radioprotective effects *in vitro*.** (A) Immunofluorescence of human BJs fibroblasts. Metformin 0.5 mM pre-treatment decreases the number of γH2AX foci in irradiated cells, compared to irradiated only controls. (B) Comet tail assay showing a decrease of the comet tail intensity in irradiated human BJ fibroblasts pre-treated with metformin 0.5 mM compared to the irradiated controls (student's t-test, Welch correction, **p<0.01).
(TIF)

**S4 Fig. Metformin pre-administration decreases the oxidative stress, retains antioxidant effect and decreases apoptosis in colon and lung tissues in mice irradiated with GCRsim.** (A, B) Metformin retains the expression of OGG1 and decreases cleaved PARP expression in murine colon and lung tissues. Furthermore, metformin activates AMPK.
(TIF)

## Acknowledgments

This work would not have been possible without support from Dr. Adam Rusek and Peter Guida and other members of the BNL and NSRL team.

## Author Contributions

**Conceptualization:** Silvia Siteni, Jerry W. Shay.

**Data curation:** Silvia Siteni, Jerry W. Shay.

**Formal analysis:** Silvia Siteni.

**Funding acquisition:** Jerry W. Shay.

**Investigation:** Silvia Siteni, Jerry W. Shay.

**Methodology:** Silvia Siteni, Summer Barron, Krishna Luitel.

**Project administration:** Silvia Siteni, Jerry W. Shay.

**Resources:** Summer Barron, Jerry W. Shay.

**Software:** Silvia Siteni.

**Supervision:** Silvia Siteni, Jerry W. Shay.

**Validation:** Silvia Siteni.

**Visualization:** Silvia Siteni.

**Writing – original draft:** Silvia Siteni.

**Writing – review & editing:** Summer Barron, Krishna Luitel, Jerry W. Shay.

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
