## [Decision Letter · Decision Letter 0]

8 May 2024

PONE-D-24-13104RADIOPROTECTIVE EFFECT OF THE ANTI-DIABETIC DRUG METFORMINPLOS ONE

Dear Dr. Shay,

Thank you for submitting your manuscript to PLOS ONE. After careful consideration, we feel that it has merit but does not fully meet PLOS ONE’s publication criteria as it currently stands. Therefore, we invite you to submit a revised version of the manuscript that addresses the points raised during the review process.

We look forward to receiving your revised manuscript.

Kind regards,

Hanbing Li, Ph.D.

Academic Editor

PLOS ONE

Journal Requirements:

   "This work was supported by NASA Grants # NNX16AE08G to JWS and NNX15AI21G to J.W.S and Albert Fornace. K.L. was supported by Cancer Training Grant T32CA124334. This work was performed in laboratories constructed with support from NIH grant C06 RR30414. We also acknowledge the Harold Simmons NCI Designated Comprehensive Cancer Center Support Grant (CA142543)."

   "J.W.S. holds the distinguished Southland Financial Corporation chair in Geriatrics Research. J.W.S is a SAB member of Reata Pharmaceutical, Inc. (Irving, TX).  S.S, S.B and K.L. declare no competing interests."

4. In the online submission form you indicate that your data is not available for proprietary reasons and have provided a contact point for accessing this data. Please note that your current contact point is a co-author on this manuscript. According to our Data Policy, the contact point must not be an author on the manuscript and must be an institutional contact, ideally not an individual. Please revise your data statement to a non-author institutional point of contact, such as a data access or ethics committee, and send this to us via return email. Please also include contact information for the third party organization, and please include the full citation of where the data can be found.

Additional Editor comments:

Thank you for submitting your manuscript for consideration for publication in PLOS ONE. Please address the comments raised by the reviewers and submit the revised version of your manuscript later.

Reviewers' comments:

Reviewer's Responses to Questions

**Comments to the Author**

1. Is the manuscript technically sound, and do the data support the conclusions?

Reviewer #1: Yes

Reviewer #2: Yes

2. Has the statistical analysis been performed appropriately and rigorously? 

Reviewer #1: Yes

Reviewer #2: Yes

3. Have the authors made all data underlying the findings in their manuscript fully available?

Reviewer #1: Yes

Reviewer #2: Yes

4. Is the manuscript presented in an intelligible fashion and written in standard English?

Reviewer #1: Yes

Reviewer #2: Yes

5. Review Comments to the Author

Reviewer #1: In the current manuscript, the authors have repurposed metformin for its radioprotective effects, which are currently being used as a first-line treatment for managing type 2 diabetes. The authors utilized various in vitro and in vivo assays to explore its beneficial effects, including its potential use in space missions using galactic cosmic ray simulation (GCRsim). Additionally, the authors studied the possible mechanisms of radioprotection by metformin. In my opinion, the manuscript is well-written and organized.

However, I have several queries that the authors need to address in their revised version of the manuscript:

1. In the determination of in vitro concentrations, the authors tested concentrations of 0.5, 1, 2.5, and 5 mM of metformin, ultimately deciding to use the lowest concentration of 0.5 mM. Why did they not explore even lower concentrations than 0.5 mM?

2. For the in vivo study, the authors administered 200 mg/kg b.wt of metformin. How did the authors determine this concentration?

3. What is the LD50/30 dose for 129/sv mice? I ask this because researchers generally use sub-lethal doses for mechanistic studies and slightly higher doses than the LD50/30 for survival studies. In this study, the authors used 7.5 Gy for mechanistic studies and 10 Gy for survival studies.

Reviewer #2: The manuscript presents a comprehensive study investigating the potential radioprotective effects of metformin and its underlying mechanisms. Here's a review based on the provided criteria:

Technical Soundness and Data Support for Conclusions: The manuscript appears technically sound, with a detailed exploration of the effects of metformin on various cellular processes, particularly in response to ionizing radiation. The authors provide data on AMPK activation, changes in antioxidant enzyme expression, DNA repair kinetics, and survival outcomes in both in vitro and in vivo models. The findings suggest that metformin pre-administration confers radioprotection by enhancing DNA repair mechanisms and mitigating cellular damage induced by ionizing radiation. However, while the data presented support the conclusions drawn, there are some areas where further clarification or additional experiments may strengthen the claims. For instance, it would be beneficial to include more detailed analyses of the molecular pathways involved in metformin-mediated radioprotection, such as downstream targets of AMPK activation and the interplay between different cellular processes.

Presentation and Language: The manuscript is generally well-presented and written in standard English, making it intelligible to readers. The authors provide clear descriptions of experimental procedures, results, and interpretations, facilitating understanding of the study's findings. However, there are sections where the text could be streamlined for clarity and conciseness. Additionally, the manuscript could benefit from improved organization and structure to enhance readability and flow. For example, grouping related findings together and providing more explicit transitions between sections would help guide the reader through the study's complex data and analyses.

Overall, the manuscript presents valuable insights into the potential radioprotective effects of metformin and contributes to our understanding of its mechanisms of action. With some revisions to address the points mentioned above, it has the potential to make a significant contribution to the field of radiobiology and could be considered for publication in a scientific journal.

6. PLOS authors have the option to publish the peer review history of their article (what does this mean?). If published, this will include your full peer review and any attached files.

Reviewer #1: No

Reviewer #2: **Yes: **Nur Fariesha Md Hashim

---

## [Author Response · Author response to Decision Letter 0]

22 Jun 2024

Review Comments to the Author

We want to thank the editor and the reviewers for taking the time to read our paper and for the relevant questions.

Our responses to the initial review comments are detailed in the revised manuscript and summarized below.

Reviewer #1: In the current manuscript, the authors have repurposed metformin for its radioprotective effects, which are currently being used as a first-line treatment for managing type 2 diabetes. The authors utilized various in vitro and in vivo assays to explore its beneficial effects, including its potential use in space missions using galactic cosmic ray simulation (GCRsim). Additionally, the authors studied the possible mechanisms of radioprotection by metformin. In my opinion, the manuscript is well-written and organized.

However, I have several queries that the authors need to address in their revised version of the manuscript:

1. In the determination of in vitro concentrations, the authors tested concentrations of 0.5, 1, 2.5, and 5 mM of metformin, ultimately deciding to use the lowest concentration of 0.5 mM. Why did they not explore even lower concentrations than 0.5 mM?

Thank you very much for pointing out an important question. We did test lower concentrations of metformin and we focused on the lower concentrations of metformin able to induce the activation of SOD1. We added the data in the revised submission (S1 Fig B).

2. For the in vivo study, the authors administered 200 mg/kg b.wt of metformin. How did the authors determine this concentration?

Thank you for the question. We explained our choice in the material and methods section (p. 11, Drug preparation). Briefly, we evaluated the usual metformin administration in humans (between 1000 and 2000 mg per day). We used the body surface area (BSA) normalization method [53] according to the FDA guidelines [54] and we found a minimal dosage of 250 mg/kg. However, since our aim was to study the radioprotective effects of metformin in healthy wild type mice (and potentially in very healthy human beings such as astronauts) we decreased the concentration to 200 mg/kg. We explained this better in the revised manuscript

3. What is the LD50/30 dose for 129/sv mice? I ask this because researchers generally use sub-lethal doses for mechanistic studies and slightly higher doses than the LD50/30 for survival studies. In this study, the authors used 7.5 Gy for mechanistic studies and 10 Gy for survival studies.

According to previous studies in our lab and others, wild type mice are sensitive to 7.5 Gy of total body x-rays. However, when we did a pilot test, our mice were not lethally affected by 7.5 Gy, but most wild type mice died after two weeks when irradiated once with 10 Gy x-rays. We used the sublethal dose of 7.5 Gy to better investigate the DNA repair kinetics and the lethal dose of 10 Gy to evaluate the overall survival in the presence or absence of metformin.

Reviewer #2: The manuscript presents a comprehensive study investigating the potential radioprotective effects of metformin and its underlying mechanisms. Here's a review based on the provided criteria:

Technical Soundness and Data Support for Conclusions: The manuscript appears technically sound, with a detailed exploration of the effects of metformin on various cellular processes, particularly in response to ionizing radiation. The authors provide data on AMPK activation, changes in antioxidant enzyme expression, DNA repair kinetics, and survival outcomes in both in vitro and in vivo models. The findings suggest that metformin pre-administration confers radioprotection by enhancing DNA repair mechanisms and mitigating cellular damage induced by ionizing radiation. However, while the data presented support the conclusions drawn, there are some areas where further clarification or additional experiments may strengthen the claims. For instance, it would be beneficial to include more detailed analyses of the molecular pathways involved in metformin-mediated radioprotection, such as downstream targets of AMPK activation and the interplay between different cellular processes.

Thank you for appreciating our work and paper. We agree with the reviewer about including a deeper investigation into the mechanisms to elucidate the role of AMPK effects downstream. We did investigate the main pathways involving the DNA damage response and AMPK. Below is what we discussed in the original draft that addresses this comment:

 “…we found metformin 0.5 mM increases the phosphorylation of ataxia telangiectasia mutated (ATM) at serine 1981 and p53 expression after 48 hours. ATM is a kinase that acts upstream of p53 and controls a DDR pathway critical to resolving DSBs [45]. However, we did not find any increase in γH2AX foci in vitro or in 53BP1 foci in vivo, but this might be related to the time-points we chose to analyze. Vazquez-Martin et al. [46] reported ATM phosphorylation at serine 1981 and formation of γH2AX foci after metformin exposure in epidermoid carcinoma cells, but not of 53BP1. This is in line with what was reported by Bakkenist & Kastan [47], showing ATM activation is not dependent on direct binding to DNA strand breaks, but may result from changes in the structure of chromatin. Interestingly, the increased expression of p53 might have effects on the cell cycle, inducing a transient arrest in G1 phase, and thus providing time for DNA repair from the effects of IRs.”

We did not obtain significant data about CHK1, CHK2, BRCA1, ATR expression and these results are not included in the revised manuscript.

---

## [Decision Letter · Decision Letter 1]

9 Jul 2024

RADIOPROTECTIVE EFFECT OF THE ANTI-DIABETIC DRUG METFORMIN

PONE-D-24-13104R1

Dear Dr. Jerry W Shay,

We’re pleased to inform you that your manuscript has been judged scientifically suitable for publication and will be formally accepted for publication once it meets all outstanding technical requirements.

Kind regards,

Hanbing Li, Ph.D.

Academic Editor

PLOS ONE

Additional Editor Comments (optional):

Reviewers' comments:

Reviewer's Responses to Questions

**Comments to the Author**

1. If the authors have adequately addressed your comments raised in a previous round of review and you feel that this manuscript is now acceptable for publication, you may indicate that here to bypass the “Comments to the Author” section, enter your conflict of interest statement in the “Confidential to Editor” section, and submit your "Accept" recommendation.

Reviewer #1: All comments have been addressed

Reviewer #2: All comments have been addressed

2. Is the manuscript technically sound, and do the data support the conclusions?

Reviewer #1: Yes

Reviewer #2: Yes

3. Has the statistical analysis been performed appropriately and rigorously? 

Reviewer #1: Yes

Reviewer #2: Yes

4. Have the authors made all data underlying the findings in their manuscript fully available?

Reviewer #1: Yes

Reviewer #2: Yes

5. Is the manuscript presented in an intelligible fashion and written in standard English?

Reviewer #1: Yes

Reviewer #2: Yes

6. Review Comments to the Author

Reviewer #1: Authors have taken care of all the queries raised by me and accordingly revised the manuscript. Now it can be accepted.

Reviewer #2: Thank you for addressing my comments and questions with detailed and thoughtful responses. The inclusion of additional data on lower concentrations of metformin and the rationale for the in vivo dosage selection significantly strengthens the manuscript. Your explanation of the sublethal and lethal doses used in the mouse model studies, and the addition of the data on SOD1 activation, provide greater clarity and support for your experimental design. Furthermore, your elaboration on the downstream targets of AMPK activation and the role of ATM and p53 in the DNA damage response enhances our understanding of the molecular pathways involved in metformin-mediated radioprotection. The revised manuscript is well-organized and provides comprehensive insights into the radioprotective effects of metformin, supporting its potential application in space missions and radioprotection. I recommend acceptance of the manuscript after incorporating the discussed revisions.

7. PLOS authors have the option to publish the peer review history of their article (what does this mean?). If published, this will include your full peer review and any attached files.

Reviewer #1: No

Reviewer #2: No

---

## [Editor Report · Acceptance letter]

12 Jul 2024

PONE-D-24-13104R1 

PLOS ONE

Dear Dr. Shay, 

I'm pleased to inform you that your manuscript has been deemed suitable for publication in PLOS ONE. Congratulations! Your manuscript is now being handed over to our production team.

Kind regards, 

on behalf of

Professor Hanbing Li 

Academic Editor

PLOS ONE